# Grapevine Red Blotch Disease Etiology and Its Impact on Grapevine Physiology and Berry and Wine Composition

Arran C. Rumbaugh [1], Mysore R. Sudarshana [2] and Anita Oberholster [1,*]

[1] Department of Viticulture & Enology, University of California Davis, One Shields Avenue, Davis, CA 95616, USA; acrumbaugh@ucdavis.edu
[2] United States Department of Agriculture, Department of Plant Pathology, University of California Davis, One Shields Avenue, Davis, CA 95616, USA; mysore.sudarshana@usda.gov
* Correspondence: aoberholster@ucdavis.edu

**Abstract:** Grapevine red blotch virus (GRBV) has become widespread in the United States since its identification in 2012. GRBV is the causative agent of grapevine red blotch disease (GRBD), which has caused detrimental economic impacts to the grape and wine industry. Understanding viral function, plant–pathogen interactions, and the effects of GRBV on grapevine performance remains essential to developing potential mitigation strategies. This comprehensive review examines the current body of knowledge regarding GRBV, to highlight gaps in the knowledge and potential mitigation strategies for grape growers and winemakers.

**Keywords:** grapevine red blotch virus; transmission; grape metabolism; wine chemistry; review





## 1. Introduction

Plant viruses detrimentally impact crops around the world by reducing yields or decreasing crop quality. Unlike other plant pathogens, viruses are obligate intracellular parasites that require the host's machinery to replicate. *Vitis vinifera* is one of the most susceptible plant hosts to viral infection, with over 80 viruses recorded that potentially impact grapevine performance [1]. Major grapevine viruses are associated with four main disease complexes: (i) viruses responsible for infectious degeneration or decline disease, (ii) viruses associated with leafroll disease, (iii) viruses associated with the rugose wood complex, and (iv) viruses associated with the fleck complex [2]. A vast majority of these viruses comprised an RNA genome, with DNA viruses being relatively rare. Some of the most detrimental of these viruses to the grape and wine industry are grapevine fanleaf virus (GFLV), grapevine leafroll-associated viruses (GLRaV), and the recently recognized grapevine red blotch virus (GRBV) [3–5].

Of the known GLRaVs, GLRaV-3 is the most important etiological agent of grapevine leafroll disease (GLRD) [6]. GLRaVs affect berry ripening by decreasing sugar accumulation and anthocyanin biosynthesis [7–9]. Foliar symptoms include interveinal reddening, with the veins remaining green in red cultivars, with the interveinal area of leaves of white cultivars becoming chlorotic. Currently, no sources of resistance to GLRaVs have been documented in *V. vinifera* cultivars or clones [6,10,11]. However, variable responses to GLRaV infection has been recently reported, with some rootstocks outperforming others [9].

In 2008, Cabernet Sauvignon grapevines in Oakville, California (Oakville Experimental Station, Napa County, CA, USA) were noticed to have symptoms that resembled leafroll disease. However, in laboratory tests, symptomatic vines tested negative for all known leafroll viruses; thus, this new disease was termed grapevine red blotch disease (GRBD). Simultaneously, in New York, Oregon and Washington state, other researchers experienced the same phenomena. Independently, these research groups used rolling circle amplification (RCA) or large-scale sequencing methods to identify a new circular ssDNA virus comprising 3206 nt [12–14]. During this time, multiple nomenclatures were used to identify

this virus: grapevine cabernet franc-associated virus [12], grapevine red blotch-associated virus [13], and grapevine geminivirus [14]. The almost identical isolates in these studies indicated that the same virus was infecting grapevines in multiple states across the United States [15,16], and the name grapevine red blotch-associated virus (GRBaV) was retained. Subsequently, GRBaV was included in the family Geminiviridae family of viruses and was found to be the causative agent of GRBD [12,13,17]. Therefore, the name grapevine red blotch virus (GRBV) was adopted and will be utilized for the remainder of this review. Since its identification, GRBV presence has been reported in vineyards worldwide [18–22] and in raisin and table grapes [23] (Table 1). Interestingly, the presence of GRBV has remained absent in Old World vineyards [24].

**Table 1.** Distribution of GBRV in the US and around the world with the cultivar(s) and date reported.

| Location | Country | Cultivar | Reference |
|---|---|---|---|
| California | USA | Cabernet franc | Al Rwahnih et al. 2012, 2013 [13,25] |
| California | USA | Zinfandel | Al Rwahnih et al. 2012, 2013 [13,25] |
| New York | USA | Cabernet franc | Krenz et al. 2012 [12] |
| Washington | USA | Merlot | Poojari et al. 2013 [26] |
| Washington | USA | Cabernet franc | Poojari et al. 2013 [26] |
| Texas | USA | Unknown | National Clean Plant Network 2013 [27] |
| Pennsylvania | USA | Merlot | Krenz et al. 2014 [15] |
| Pennsylvania | USA | Cabernet franc | Krenz et al. 2014 [15] |
| New York | USA | Pinot noir | Krenz et al. 2014 [15] |
| California | USA | Chardonnay | Krenz et al. 2014 [15] |
| California | USA | Pinot noir | Krenz et al. 2014 [15] |
| California | USA | Cabernet Sauvignon | Krenz et al. 2014 [15] |
| California | USA | Malbec | Krenz et al. 2014 [15] |
| California | USA | Petit Verdot | Krenz et al. 2014 [15] |
| California | USA | Cabernet franc | Krenz et al. 2014 [15] |
| California | USA | Riesling | Krenz et al. 2014 [15] |
| California | USA | Zinfandel | Krenz et al. 2014 [15] |
| Maryland | USA | Merlot | Krenz et al. 2014 [15] |
| Maryland | USA | Cabernet franc | Krenz et al. 2014 [15] |
| Virginia | USA | Unknown | Krenz et al. 2014 [15] |
| New Jersey | USA | Cabernet franc | Krenz et al. 2014 [15] |
| Oregon | USA | Pinot noir | Krenz et al. 2014 [15]; Seguin et al. 2014 [14] |
| California (herbarium) | USA | Early Burgundy | Al Rwahnih et al. 2015 [28] |
| California (National Clonal Germplasm Repository) | USA | Table grapes | Al Rwahnih et al. 2015 [23] |
| Arkansas | USA | Unknown | Sudarshana et al. 2015 [16] |
| Unknown | USA | Chambourcin (interspecific hybrid) | Sudarshana et al. 2015 [16] |
| California | USA | Free-living *Vitis* spp. | Perry et al. 2016 [29] |
| California | USA | Free-living *Vitis* spp. | Bahder et al. 2016 [30] |
| Suwon and Gyeongsan | South Korea | Unknown | Lim et al. 2016 [18] |
| Ontario | Canada | Cabernet franc | Poojari et al. 2017 [22] |
| Ontario | Canada | Chardonnay | Poojari et al. 2017 [22] |
| Ontario | Canada | Riesling | Poojari et al. 2017 [22] |
| Ontario | Canada | Cabernet franc | Poojari et al. 2017 [22] |
| Ontario | Canada | Syrah | Poojari et al. 2017 [22] |
| British Columbia | Canada | Muscat | Poojari et al. 2017 [22] |
| British Columbia | Canada | Cabernet franc | Poojari et al. 2017 [22] |
| British Columbia | Canada | Chardonnay | Poojari et al. 2017 [22] |
| British Columbia | Canada | Zinfandel | Poojari et al. 2017 [22] |
| British Columbia | Canada | Grenache | Poojari et al. 2017 [22] |
| British Columbia | Canada | Petit Verdot | Poojari et al. 2017 [22] |
| Nyon (Agroscope grapevine virus collection) * | Switzerland | Gamay | Reynard et al. 2018 [24] |

**Table 1.** *Cont.*

| Location | Country | Cultivar | Reference |
|---|---|---|---|
| Georgia | USA | Cynthiana (Norton, interspecific hybrid) | Brannen et al. 2018 [31] |
| Georgia | USA | Cabernet franc | Brannen et al. 2018 [31] |
| Missouri | USA | Crimson Cabernet | Schoelz et al. 2018 [32] |
| Ontario | Canada | Cabernet Franc | Xiao et al. 2018 [33] |
| Ontario | Canada | Cabernet Sauvignon | Xiao et al. 2018 [33] |
| Ontario | Canada | Pinot noir | Xiao et al. 2018 [33] |
| Ontario | Canada | Merlot | Xiao et al. 2018 [33] |
| Ontario | Canada | Syrah | Xiao et al. 2018 [33] |
| Ontario | Canada | Pinot Gris | Xiao et al. 2018 [33] |
| Ontario | Canada | Sauvignon Blanc | Xiao et al. 2018 [33] |
| Ontario | Canada | Chardonnay | Xiao et al. 2018 [33] |
| Ontario | Canada | Riesling | Xiao et al. 2018 [33] |
| Ontario | Canada | Gewürz traminer | Xiao et al. 2018 [33] |
| San Juan and Mendoza | Argentina | Flame Seedless | Luna et al. 2019 [20] |
| Baja California and Ensenada | Mexico | Pinot noir | Gasperin-Bulbarela et al. 2019 [19] |
| Baja California and Ensenada | Mexico | Merlot | Gasperin-Bulbarela et al. 2019 [19] |
| Baja California and Ensenada | Mexico | Nebbiolo | Gasperin-Bulbarela et al. 2019 [19] |
| Punjab | India | Unknown | Marwal et al. 2019 [21] |
| Tennessee | USA | Several cultivars | Soltani et al. 2020 [34] |
| Quebec | Canada | Pinot noir | Fall et al. 2020 [35] |
| Nova Scotia | Canada | Chardonnay | Poojari et al. 2020 [36] |
| Nova Scotia | Canada | Pinot noir | Poojari et al. 2020 [36] |
| Nova Scotia | Canada | New York Muscat (Interspecific hybrid) | Poojari et al. 2020 [36] |
| Nova Scotia | Canada | Marechal Foch (Interspecific hybrid) | Poojari et al. 2020 [36] |
| Idaho | USA | Syrah | Lee et al. 2021 [37] |

\* GRBV was reported absent in commercial Switzerland vineyards.

Currently, an increasing number of new geminiviruses are being discovered, most likely due to the increasing capabilities of high-throughput sequencing technologies. Due to globalization and exchanging of planting material, geminiviruses are rapidly expanding internationally and infecting several different hosts, causing new diseases and epidemics. Grape and wine production is one of the most economically important industries globally. With the economic impact of GRBV ranging from 2213 USD/ha to 68,548 USD/ha in the United States [4], recent research has focused on virus functioning, epidemiology, impact on grape metabolism, and wine quality, as well as mitigation strategies. This review examines the existing body of knowledge regarding the viral genome, virus transmission, and the impacts of GRBV on grapevine physiology, grape metabolism, and wine composition. Sensory analysis of wine made from GRBV-infected fruit is also discussed. Due to the impact of GRBV on grape and wine composition, recent research has revealed potential viticultural and enological mitigation strategies. Although great advancement in our knowledge of GRBV has been achieved, several important research questions remain unanswered and are discussed here.

## 2. GRBV Genome and Taxonomy

The first group to identify GRBV used deep sequencing of dsRNA fractions extracted from symptomatic grapevines followed by RCA on total nucleic acid extracts [25]. Through sequencing of RCA product, the circular monopartite ssDNA virus was identified. Phylogenetic analyses of the coat protein and replicase-associated protein sequences revealed GRBV to group with the family Geminiviridae [12,13,15,25]. However, this was outside all seven of the recognized genera of the time. At the time of its discovery, GRBV was the second largest geminivirus genome, with 3206 nt, and the closest related sequence, only sharing 50% identity, was a dicot-infecting *Mastrevirus*, chickpea chlorotic dwarf Syria

virus [12,15]. In 2017, a new genus, *Grablovirus*, was established with GRBV as the type species [38]. The genus *Grablovirus* now includes two new viruses: wild Vitis virus 1 and Prunus geminivirus A [39–41].

The GRBV genome contains the characteristic nonanucleotide sequence ('TAATATT | AC') that functions as the viral origin of replication and is found in almost all members of Geminiviridae [12,13,15,16,25,26]. Like all geminiviruses, GRBV contains bidirectional open reading frames (ORFs). For GRBV, there are three virion-sense ORFs and three complementary-sense ORFs. Virion-sense ORF V1 was determined to be the coat protein, and V2 and V3 are putative movement proteins. In the complementary-sense, C1 and C2 show similarity with other mastreviruses, including a putative spliced transcript. The C1 and C2 spliced transcript is thought to encode for the replication protein (Rep) (Figure 1). C3 is in the same reading frame as C1 and is internal. However, more recently, research has uncovered a seventh ORF, V0, a small ORF upstream of V2, also thought to be associated with viral movement [39]. This second splicing event in the virion-sense was discovered through investigating evidence for C1 and C2 splicing. Although virion-sense splicing is rarer than complementary-sense splicing for geminiviruses, it does occur in mastreviruses and capulaviruses (both in the Geminiviridae family) [39,42,43]. The occurrence in GRBV is a proposed regulatory enhancement to V1 gene expression due to the arrangement of V0, V2, and V1 [39].

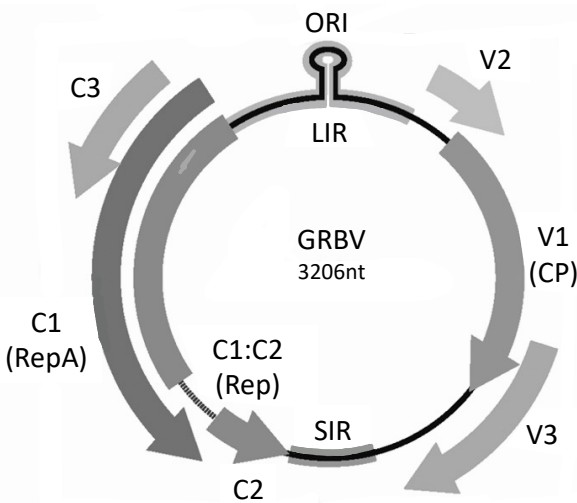

**Figure 1.** Genome organization of the genera Grablovirus. Adapted from International Committee on Taxonomy of Viruses [44].

Phylogenetic analysis on genomes of known isolates of GRBV two distinct clades: clade 1 and clade 2. Clade 1 was determined to have the highest variability, at 94.8% [15]. By comparing the GRBV genomes, recombination was associated with some of the variation observed that could influence the evolution of GRBV and may potentially contribute to the emergence of new virus variants [15,22]. Isolates in clade 2 showed less variability, at 98.8%, and contained the majority of the isolates analyzed. Between the two clades, nucleotide identity ranges from 91–93% [16]. Analysis of historical specimens of California revealed that a specific PCR product shared 97–100% nucleotide homology with GRBV. This specimen was collected from Sonoma County in 1940 and shared close nucleotide identity with clade 2 [28], indicating the presence of GRBV much earlier than 2008.

## 3. Causative Role in GRBD: Symptoms, Diagnosis, and Transmission

Many grapevine viruses, besides GFLV [45], have not been identified as the causal agent of their associated diseases. Although GRBV was associated with GRBD, it was not until 2018 that its etiological role in GRBD was proven. Through engineering infectious GRBV clones and agroinoculation, all four of Koch's postulates were fulfilled [17], thus establishing GRBV the causative agent of GRBD.

Symptoms of GRBD consist of red blotches on the leaf blades and margins, with reddening of the primary, secondary, and tertiary veins in red berry cultivars (as seen in Figure 2). In white berry cultivars, the foliar symptoms are less conspicuous and generally involve chlorotic lesions [46]. Foliar symptoms are not reported to appear until after veraison, with mature basal leaves being more symptomatic than the middle and terminal leaves and eventually dropping off prematurely when heavily symptomatic. The virus has also been detected in the roots, fruit clusters, and fruit juice [13,16]. Due to the similarity to abiotic and biotic stressors, such as nutrient deficiencies and other diseases, the most accurate method to diagnose GRBV is DNA-based assays. However, another approach was developed using mass spectrometry to quantify GRBV in infected plants [47]. This report was the first to physically detect the predicted V1 and V2 gene products at the protein level. Based on the AAFNIFQR peptide abundance, the coat protein was consistently identified in higher amounts in petiole extracts of GRBV-infected plants compared to leaf extracts.

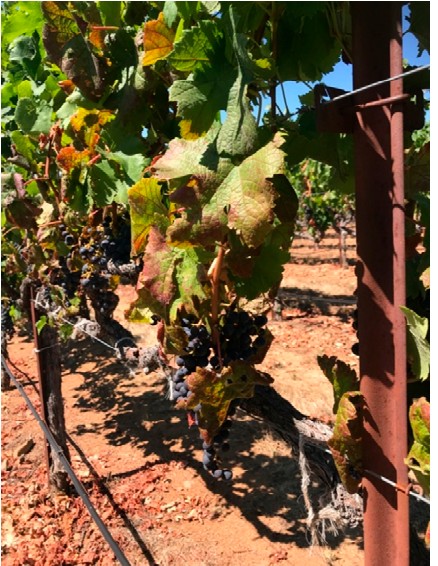 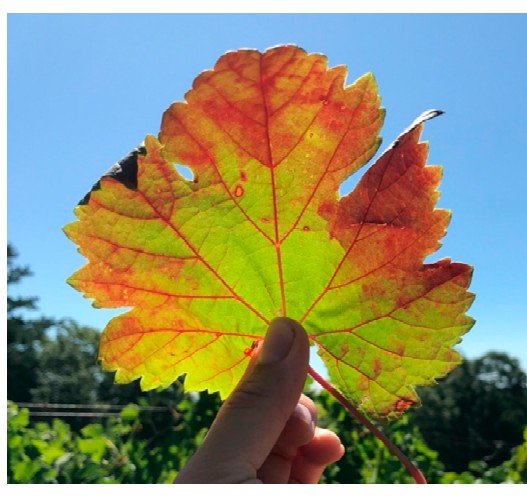

**Figure 2.** Foliar symptoms of GRBD in Merlot grapevines in Napa, CA, USA. Photo credit: Arran Rumbaugh.

The extent of the viral spread of GRBV in a vineyard depends on the location in North America. In New York, secondary spread via an insect vector has not been reported, and over a four-year study, no new infections were found [48]. However, other studies reported between 1–14% spread within a season in California vineyards [48,49], 11–55% in Oregon vineyards in two seasons [50], and 1% spread in British Columbia vineyards within a season [22]. Therefore, several researchers set out to determine the leading cause of new infections within a vineyard.

GRBV is a graft-transmissible, phloem-limited virus, with systemic movement detected in leaves distal to the graft site [13,26]. This, and the fact that GRBV infects several varieties, suggests that propagation material is the primary method of viral spread in the United States. However, there is also evidence of viral spread caused by insect vectors. Research in the past five years on the spatiotemporal analysis of viral spread identified new GRBV infections near the vineyard's edge proximal to riparian habitats [49–52]. In addition, GRBV has been detected in free-living vines proximal to vineyards [29,30] but has yet to be detected in cover crops [48]. These results are consistent with short-distance spread of the virus potentially from a flying insect vector. Previous research indicates that GRBV is closely related to geminiviruses transmitted by *Auchenorrhyncha*, which are leafhoppers and treehoppers [51]. One of the first identified vectors in greenhouse settings was the Virginia creeper leafhopper, *Erythroneura ziczac* [26]. However, this insect mainly

feeds on the mesophyll, not the phloem of plants, and GRBV was not reported to spread in regions of North America where *E. ziczac* is well-established [51].

The current recognized vector of GRBV is the three-cornered alfalfa hopper (*Spissistilus festinus,* Membracidae), yet successful transmission to grapevines via *S. festinus* has only been achieved in greenhouse settings [51]. Cieniewicz et al. (2017a) corroborated these results in which *S. festinus* was the only insect vector to show significant associations with the spatial pattern of infected vines. Higher numbers of *S. festinus* were found near the vineyard edges next to riparian areas, associating these habitats as potential infection sites for GRBV. Additional studies demonstrated circulative, nonpropagative transmission of GRBV by *S. festinus*, where the insect was able to successfully transmit GRBV to grapevine leaves [53]. However, other studies reveal a dissociation between *S. festinus* presence and viral spread in a vineyard, suggesting another vector may transmit GRBV [50]. To date, successful GBRV transmission by another insect has not been proven; yet, GRBV was detected in *Osbornellus borealis* (Cicadellidae), *Colladonus reductus* (Cicadellidae), and a *Melanoliarus* sp. (Cixiidae) [52], as well as *Stictocephala bisonia* and *Stictocephala basalis* [54], making them potential vector candidates.

## 4. Impacts on Grapevine Physiology

Many plant viruses cause reductions in yields as well as decreases in crop quality. Grapevine viruses are no different, with many viruses detrimentally affecting the grape and wine industry. However, GRBV is the first identified geminivirus to infect grapevines, and its discovery was largely a result of poor juice wine quality in grapevines not known to have any leafroll-associated viruses. Since then, numerous studies have described the effects on grapevine performance in grapevines found to be infected by GRBV. Reports generally indicate a reduction in winter pruning weights, crop yield, as well as a change in berry weight. Pruning weights are consistently lower in GRBV-infected vines, with reductions ranging from 20–35% for GRBV-infected vines compared to healthy vines, suggesting that GRBV decreases vine vigor [24,26,46,55].

In Washington vineyards, crop yield decreased in GRBV-infected Merlot and Cabernet Franc vines by 22% and 37%, respectively, which the authors attributed to a lower number of clusters per vine [26]. Similarly, in 2020, a 42% reduction in crop yield was reported, with 19% fewer clusters per vine and 47% fewer berries per cluster due to GRBV infections in Cabernet Franc grapevines in British Colombia vineyards [55]. White-berried cultivars exhibited similar reductions in crop yields, with infected vines having as much as 22% lower yields compared to healthy vines [46]. Nonetheless, these results were inconsistent with data collected from Cabernet Sauvignon grapevines in California and Syrah grapevines in Idaho, where no significant differences were observed for crop yield and pruning weights [37,56]. Interestingly, increases in berry mass were also reported [46,55–57], which likely was caused by increased space due to fewer berries per cluster.

Decreased yields are potentially associated with decreased bud hardiness, photosynthesis, and stomatal conductance due to GRBV infection [24,55,56,58,59]. In healthy grapevines, higher sugar concentrations in the leaves due to decreased transportation through phloem network into sinks (i.e., berries) can suppress photosynthesis [60]. When photosynthate production exceeds the translocation of hexoses, namely sucrose, from source to sink, surplus sucrose is transported to the guard cells resulting in stomatal closure [61]. Virus-infected leaves are known to have decreased photosynthesis and increased respiration and photosynthate products (i.e., sucrose), suggesting that viral infections can alter source-to-sink pathways in infected plants, where the leaves function as sinks. Higher foliar sugar levels have been reported in GRBV-infected grapevines [26,56,59], which is similar to the leaves of sugar beets infected by beet curly top virus, a monopartite virus known to affect sugar beet production in the US for over a century [62]. Martínez-Lüscher et al. [56] proposed that GRBV impairs the translocation of sucrose from source to sink (leaves to fruit), resulting in decreases in stomatal conductance, better plant water status (stem water potential), and eventually leading to increases in foliar sugar levels. However,

physical impairment of phloem unloading through callose deposits or other processes has not been observed. Levin and KC [57] proposed a similar picture on the seasonal progression of GRBD symptom development, and suggest that reduction in stomatal conductance and leaf gas exchange and the onset of red-leaf foliar symptoms precedes the increase in stem water potential. Additionally, their data showed that the onset of foliar symptoms were not dependent on water status changes but on other factors such as the carbohydrate/nutrient alterations proposed by Martinez et al. [56].

Examination of foliar metabolite concentrations revealed higher concentrations of phenolic levels [59], with decreases in chlorophyll a and b and carotenoid concentrations [24,55,58], all of which relate to premature senescence due to GRBV infection. Specific amino acid concentrations were also higher in GRBV-infected leaves. Glycine, lysine, and proline were found to be consistently higher through grape development in two cultivars [59]. A typical plant defense mechanism to stress is the accumulation of proline. Pathogen infections were shown to activate the biosynthesis of proline via similar signaling components to salicylic acid (SA) [63], the latter also being related to plant defense responses and elevated in concentration due to GRBV infections [64]. A more in-depth examination of the proteome of GRBD-infected leaves clearly revealed higher expression of proteins than in healthy plants. Key enzymes in the phenylpropanoid pathway, ANS, ANR, and CHS, were all upregulated in GRBV-infected leaves and petioles [47]. GLRaV infections generate similar responses at the transcriptomic level, leading to the development of red foliar symptoms of GLRD [8], which are postulated to be associated with increased foliar sugar levels [26]. Together, the induction of the flavonoid pathway and increases in proline levels in GBRV-infected leaves indicates the activation of defense mechanisms.

## 5. Impairment to Grape Metabolism

Like GLRD, GRBD characteristically decreases total soluble sugars (TSS) in grape berries, supporting the notion that GRBV infection impairs the translocation of sugar from the leaves to the grape berry. Concurrently, titratable acidity (TA) and malic acid levels are higher, consistent with a disruption in grape-ripening events [16,24,37,46,55,56,58,65,66]. At veraison, energy utilization in the grape switches from sugar to organic acids, primarily malic acid. As sugars begin to accumulate in the vacuole, malic acid is transported into the cytosol and becomes available for energy metabolism, amino acid interconversions, and secondary metabolite synthesis, such as flavonoids. Malic acid catabolism results in a decrease in berry TA and increases pH. Interestingly, higher titratable acidity or malic acid content almost never correspond with lower pH values [46,47,55,58,65]. Higher potassium levels may cause this dissociation, yet only one study observed elevated potassium levels due to GRBV [46], whereas another observed decreases [67]. It should be noted that in these studies, measurements were performed on a composite grape sample with no replications from asymptomatic and symptomatic grapevines at harvest in only one season. Like sugar, potassium is also imported into the berry through the phloem. Since sucrose is transported in plants from source to sink via specific sucrose carriers (SoSUT1) [68], it is plausible that physical phloem impairment may inhibit the transport of sucrose, but not small ions. The positive correlation with potassium concentration and a plants resistance to pathogens is well-documented [69–71]. In addition, potassium concentrations affect hormone abundances of SA and jasmonic acid, which are positively related to acquired systemic resistance to pathogens [69]. In a study evaluating genetic modulation and hormonal network alterations due to GRBV infection, SA concentrations were significantly higher towards the end of ripening [64]. Although it is plausible that GRBD may lead to higher berry potassium concentrations to fight off the infection, in less than half of viral infections studied did potassium increases lead to resistance [70]. Future studies would need to investigate the ionome of grapevines to unravel the interplay between potassium and other minerals and GRBV infection. Lastly, GRBV mainly elevates berry amino acid concentrations, hypothesized to be from a reallocation of substrates for grape energy metabolism and as a defense response [65].

During berry development, many secondary metabolites are synthesized, the majority of which are highly affected by environmental and genotypic factors. Flavonoids, synthesized via the phenylpropanoid pathway, are the most widely studied due to their important organoleptic properties [72–75]. Plant–pathogen interactions derived from viral infections commonly alter flavonoid biosynthesis [64,76,77]. GRBD imparts variable alterations to flavonoid concentrations in berries, with the most damaging being decreases in anthocyanin concentrations [16,26,37,55,56,64–66,78]; however, these results are not always statistically significant. Monomeric malvidin derivatives, the most common anthocyanin form, were found to be either higher or unimpacted due to GRBV infection [56,65], to the detriment of the less abundant anthocyanin forms. Reduction in anthocyanin accumulation in grapes has been associated with genetic suppression of the phenylpropanoid pathway (68% of genes) and decreases in abscisic acid levels due to GRBV infection [64]. Abscisic acid is an essential hormone that positively regulates ripening in grapevines, and its accumulation correlates with anthocyanin biosynthesis [79,80]. Taken together, GRBV unfavorably alters the phenylpropanoid pathway, consistent with delays in ripening events.

GRBD generally increased flavonol concentrations in white-berry cultivars [46,65], potentially related to lower vine vigor increasing sun exposure [72]. However, one study did observe lower flavonol levels in grapes [55]. The concentrations of flavan-3-ols and proanthocyanidins (condensed tannins) greatly depended on the grapevine genotype and environmental/seasonal factors [46,55,56,65,78]. However, skin proanthocyanidins were occasionally higher in GRBV-infected grapes, potentially caused by a plant defense response [77].

Volatile compounds synthesized in the grape berry prior to harvest are also impacted due to GRBV infection [78]. In a two-year study evaluating the impact of GRBV on Cabernet Sauvignon grafted onto two different rootstocks, GRBV consistently decreased levels of almost all volatile compounds, except for C6 alcohols and aldehydes. These aroma compounds are synthesized in the lipoxygenase pathway and accumulate in grapes until the TSS reaches around 18° Brix [81], with the majority of them decreasing thereafter. The impact of GRBV on the volatilome of grapes further supports evidence that the virus infection delays ripening events. Similar to other reports, the extent of these effects depend on the genotypic and environmental differences [78]. The summary of GRBV impact on grapevine physiology and grape metabolism is shown in Figure 3 below.

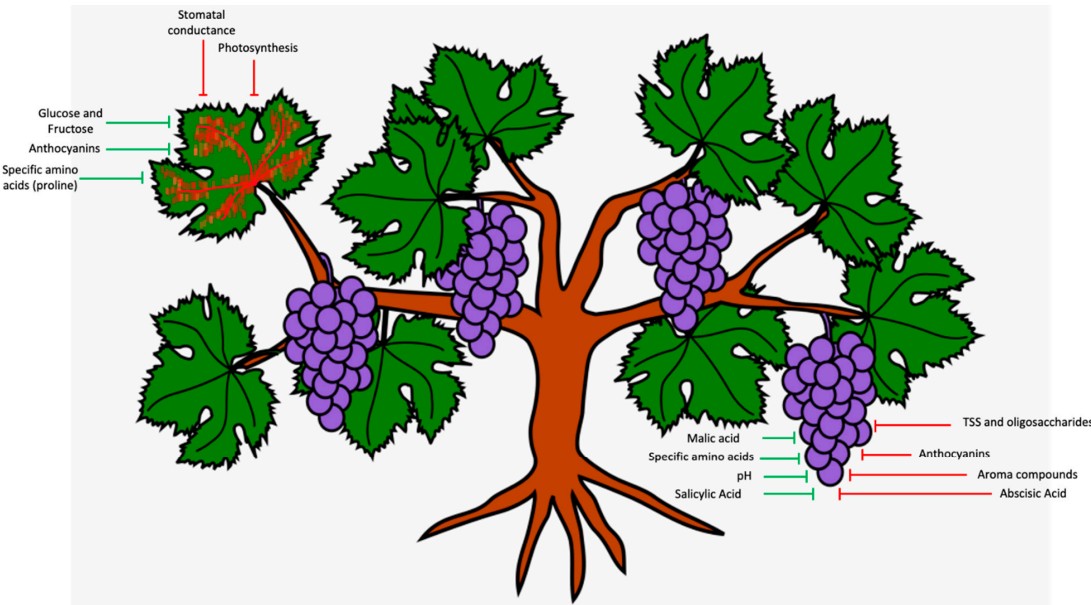

**Figure 3.** The overall impact of grapevine red blotch virus (GRBV) on grapevine physiology and grape metabolism. Green indicates an increase and red indicates a decrease in concentration. TSS = total soluble solids.

## 6. Impact on Wine Composition

Currently, very few studies report how GRBV-infected grapevines affect final wine composition and sensory attributes. Girardello et al. [46] not only analyzed the impact of GRBD on Chardonnay grape composition but also on wine composition over three seasons. Consistently, GRBV-infected vines produced wines with reduced ethanol, correlated to the lower sugar levels at harvest. Although the TA and pH of fermenting wines were adjusted in each season using tartaric acid, the pH was generally higher in wines made from diseased fruit. Once again, the higher pH was explained by higher potassium levels, where wines made from healthy fruit were 50% lower in potassium concentrations than wines made from diseased fruit [46]. However, this was only evaluated in one season, so inferences of the impact of GRBV on potassium need further confirmation. The lower ethanol content in wines made from GRBV infected grapes also seemed to affect the aroma profile of the final wines. Esterification during winemaking involves reactions of carboxylic acids and ethanol [82] to produce ethyl ester aroma compounds described as fruity and sweet (http://www.flavornet.org) (accessed on 26 July 2021). As previously reported, GRBD lowers concentrations of carboxylic acids in fruit [65] and final ethanol concentrations in wines, ultimately decreasing the production of ethyl esters in the final wines. Sensorily, these panelists were able to distinguish between Chardonnay RB(+) wines and RB(-) wines, where wines made from diseased fruit were rated significantly lower than wines made from healthy fruit for hot mouthfeel, spicy and citrus aroma, and sweet taste attributes, and significantly higher for greener aromas such as apple.

Similar results were obtained in studies analyzing the impact of GRBV on final wine composition made from red-berried cultivars. Alcohol levels were consistently lower in wines made from GRBV-infected fruit, which led to noticeable differences in the sensory characteristics [55,67]. Sour and green aromas were general attributes for wines made from GRBD fruit, correlating with unripe fruit. Simultaneously, these wines were rated lower for alcohol aroma, fruit aroma, and hot mouthfeel. Generally, the flavonoid grape composition differences were transferred into the resulting wines, where GRBV-infected fruit produced red wines that were lighter and brighter (based on analysis of wine lightness, chroma and hue values) and more astringent [55]. These differences were attributed to the reported lower anthocyanin and polymeric pigment concentrations and higher tannin concentrations, respectively [67]. It is well-accepted that flavonoid concentrations can significantly impact the overall quality of a wine, especially red wines. A research study that investigated the relationship between grape composition and perceived wine 'quality' found grapes with increased anthocyanin and skin tannin concentrations resulted in wines with increased tannin and color and better ratings by wine judges [83]. This suggests that GRBV not only detrimentally impacts grapevine performance but also wine composition and quality.

## 7. Discussion

Geminiviruses have been causing detrimental impacts to crop production and vitality for over 100 years, yet it was only in 1995 when Geminiviridae family was established [84]. GRBV is one of the newest geminiviruses identified, and is widespread throughout the United States and Canada, currently affecting premium wine-producing states. The adverse impacts of GBRV on grapevine performance, berry metabolism, and final wine composition have highlighted the importance of clean propagation material. A PCR test for GRBV has helped to identify propagation material free from GRBV before sale. Researchers are pursuing studies to further investigate virus functioning, plant–pathogen interactions, as well as transmissibility of GRBV. Determining potential insect vectors of GRBV is crucial for pest management and to impede the spread of the virus. With an increase in studies providing more information regarding GRBV, and the identification of an insect vector, *S. festinus*, an updated economic impact assessment can be made which will likely be more prominent than previously reported [4]. Mitigation strategies available to grape growers

and winemakers are limited, with rogueing infected vines or complete vineyard block replacement when the disease incidence is high (>30%) being the most reliant [4].

Few studies have attempted to examine viticultural and enological techniques that could potentially alleviate the damaging impact of GRBV on a vineyard and winery [55,56]. One study extended the ripening time of GRBV-infected fruit which further decreased TA levels and anthocyanin concentrations. It was concluded that a delayed harvest is not sufficient to coalesce all grape composition parameters, and results are unpredictable from season to season [56]. However, longer hang time and higher sugars does negate the impact of alcohol differences. Additionally, a later study examined the impact of water deficits on Pinot Noir fruit quality in GRBV-infected grapevines [57]. Authors determined that although water deficits did not impact the onset of grapevine foliar symptoms, there was an increase in symptom progression through grape ripening if the water deficits were severe. The adverse effects of water deficit on yield parameters (specifically berries per cluster) in GRBV-infected vines also indicated that GRBV may impair carbohydrate partitioning to reproductive organs during water deficits. Overall, this research concluded that in some cases, water deficit may worsen fruit quality, and that the negative impacts of GRBV on grapevine physiology and grape metabolism cannot be alleviated by water-deficit irrigation.

Alternatively, Bowen et al. [55] evaluated an enological mitigation technique to ameliorate the impact of GRBV on wine composition. They observed that small percentages of GRBV fruit included during winemaking increased the chemical and sensorial similarity to wines made from healthy fruit. However, once 20% of GRBD fruit was incorporated, the differences were noticeable and more similar to wines made from 100% GRBD fruit [55]. These findings will depend on GRBV's impact in a specific season, as large seasonal variability has been observed [65,67].

Together, these studies show the possibilities to mitigate GRBV effects available to grape growers and winemakers after GRBV is established in a vineyard. However, it is well-documented that the impact of GRBV on grapevine performance and grape metabolism is dependent on genotypic and environmental factors [37,46,55,56,65,78]. To determine potentially resistant or susceptible genotypes and favorable seasonal factors, further research is needed to examine how plant–pathogen interactions may vary.

Besides sugar content, one of the most damaging impacts of GRBV on fruit and wine quality is phenolic composition. Many factors may influence the flavonoid concentrations in a final wine, such as interactions with cell walls, cell integrity and thickness, and initial grape flavonoid concentrations. Generally, the extractability of flavonoids into the wine matrix increases as the grape matures [85]. This is due primarily to changes in grape cell-wall composition and integrity. However, there is limited research on overall plant–virus interactions regarding fruit skin cell-wall metabolism, even though the cell wall plays a crucial role in the initiation of virus spread and as a defense mechanism [86]. It was postulated that GRBV alters the cell-wall rigidity of leaves due to the increased yields of extracted proteins [47]; however, alterations to grape cell-wall compositions are still unknown. Examining how GRBV impacts grape cell-wall metabolism could lead to enological techniques to alter grape musts' composition and increase phenolic extractability and composition in a final wine.

Overall, the current body of knowledge on GRBV has dramatically expanded since 2012. Significant progress has been made in determining the impact of GRBV on grape metabolism and how this relates to wine composition and sensory characteristics. This has guided future research to understand further the viral impacts on specific metabolic pathways and plant defense mechanisms, to develop mitigation strategies.

**Author Contributions:** Conceptualization, A.C.R., M.R.S. and A.O.; writing—original draft preparation, A.C.R.; writing—review and editing, A.C.R., M.R.S. and A.O.; supervision, A.O.; project administration, A.O.; funding acquisition, A.O. All authors have read and agreed to the published version of the manuscript.

**Funding:** This review received no external funding.

**Acknowledgments:** The authors would like to thank their respective funding bodies: American vineyard foundation (AVF) and the California Department of Food and Agriculture (CDFA). The authors would also like to thank the support of the Viticulture and Enology Department, the Plant Pathology Department, and the Agricultural and Environmental Chemistry Graduate Group at the University of California, Davis, CA, USA. The work on this manuscript was partly supported by USDA-ARS CRIS project number 2032-22000-016-00D. Mention of a trademark, proprietary products, or vendor does not constitute guarantee or warranty of the product by the USDA, and does not imply its approval to the exclusion of other products and vendors that might also be suitable and the United States Department of Agriculture (USDA).

**Conflicts of Interest:** The authors declare no conflict of interest.

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
