# Peer review of "Grapevine Red Blotch Disease Etiology and Its Impact on Grapevine Physiology and Berry and Wine Composition"

_horticulturae, doi:10.3390/horticulturae7120552_

Round 1

Reviewer 1 Report

Review of horticulturae-1484544

The authors proposed a review paper on a very important grapevine viral disease namely Grapevine red blotch disease.

 The paper is certainly on-topic for the journal. It provides useful information for readers. I recommended to accept the manuscripts after consideration of the minor comments and suggestions that mentioned in the attached pdf file.

Author Response

Please see the attached PDF where each of your comments was adressed. In addition, the new manuscript version has comments to adress your concerns. 

Regards,

Dr. Anita Oberholster and Ms. Arran Rumbaugh

Reviewer 2 Report

GRBV has became a economically important virus in USA and other countries since it was found in 2012. There have been  major studies involved with molecular characterization, transmission, the effects on grape and wine about GRBV. The review is good and timely summaryof GRBV research results.

  1. Line 25-26, you can replace the reference and relative description with a new recent study, because the numbers of grapevine viruses exceed 80 species.
  2. Line 26: 'Grapevine viruses' change to 'Major grapevine viruses' .
  3. Line 38: "no sources of resistance to GLRaVs" need a referece to suport this point.
  4. Line 108: "the two isolates" , what two isolates represented? two kinds of isolates?
  5. Line 115:  How do you think GFLV have identified its etiological role?
  6. Line 184: change to '.....in GRBV-infected Merlot and Cabernet France vines.....'
  7. Line 196-197:  which status do you mean  there were higher sugar concentrations? GRBV-uninfected or infected grapevine leaves?
  8. Please check the references. Reference 71 need adding the Journal name.

Author Response

The recommended alterations have been made to this manuscript. There are comments in the margins regarding each alteration. In detail:

Line 25-26, you can replace the reference and relative description with a new recent study, because the numbers of grapevine viruses exceed 80 species.

New reference cited.

Line 26: 'Grapevine viruses' change to 'Major grapevine viruses' .

Altered.

Line 38: "no sources of resistance to GLRaVs" need a referece to suport this point.

Reference added.

Line 108: "the two isolates" , what two isolates represented? two kinds of isolates?

We apologize for this mistake. The text was meant to read "clades" not "isolates". This has been changed.

Line 115:  How do you think GFLV have identified its etiological role?

The paper that described the fulfillment of Koch's postulates is provided now in this line.

Line 184: change to '.....in GRBV-infected Merlot and Cabernet France vines.....'

Changed.

Line 196-197:  which status do you mean  there were higher sugar concentrations? GRBV-uninfected or infected grapevine leaves?

This line was meant to describe what occurs in grapevines that are pathogen-free. 

Please check the references. Reference 71 need adding the Journal name.

References have been checked. 

Regards,

Dr. Anita Oberholster and Ms. Arran Rumbaugh

Reviewer 3 Report

The MS " Grapevine red blotch virus and its impact on grape berry physiology and wine composition" is interesting, I did not find inappropriate place.  
Could you present the grapevine red blotch virus in the text? 

Author Response

We have in Figure 1. the genome organization of grapevine red blotch virus (GRBV) as well as the symptoms it causes in grapevines in Figure 2. The text also describes in detail the etiology of grapevine red blotch disease (GRBD) and the impact of GRBV on grapevine physiology, grape metabolism, and wine composition. If this is not enough to present GRBV in the text, please let us know.

Regards,

Dr. Anita Oberholster and Ms. Arran Rumbaugh